# VeLoRA: Memory Efficient Training using Rank-1 Sub-Token Projections

**Roy Miles**[†]   **Pradyumna Reddy**   **Ismail Elezi**[†]   **Jiankang Deng**
Huawei Noah's Ark Lab

## Abstract

Large language models (LLMs) have recently emerged as powerful tools for tackling many language-processing tasks. Despite their success, training and fine-tuning these models is still far too computationally and memory intensive. In this paper, we identify and characterise the important components needed for effective model convergence using gradient descent. In doing so we find that the intermediate activations used to implement backpropagation can be excessively compressed without incurring any degradation in performance. This result leads us to a cheap and memory-efficient algorithm for both fine-tuning and pre-training LLMs. The proposed algorithm simply divides the tokens up into smaller sub-tokens before projecting them onto a fixed 1-dimensional subspace during the forward pass. These features are then coarsely reconstructed during the backward pass to implement the update rules. We confirm the effectiveness of our algorithm as being complimentary to many state-of-the-art PEFT methods on the VTAB-1k fine-tuning benchmark. Furthermore, we outperform QLoRA for fine-tuning LLaMA and show competitive performance against other memory-efficient pre-training methods on the large-scale C4 dataset. Code: https://github.com/roymiles/VeLoRA

## 1   Introduction

Large language models (LLMs) have achieved remarkable success on a wide range of natural language processing tasks [31, 47, 2]. However, training these massive deep learning models remains computationally expensive, requiring vast amounts of data, compute, and memory resources. A key bottleneck for training or adapting these models is the large memory needed to store all the intermediate features required to compute the gradients for optimization. This makes it challenging to fully leverage the scalability and performance gains promised by larger models on currently available hardware.

Several techniques have been proposed to reduce the memory requirements, such as GaLore [54], gradient checkpointing [8], reversible backpropagation [14], parameter-efficient finetuning [18, 7], quantization [11] and activation offloading [30]. GaLore uses a low-rank projection of the gradients during training to reduce the memory footprint. Gradient checkpointing reduces the activation memory demands by recomputing the activations in the backward pass instead of storing them. While these methods are promising and lower the memory cost, they also might introduce a substantial computational overhead, are limited in their memory savings, or require specialized hardware [11]. Knowing that compute is the primary mover for the advancements in machine learning [46], it is to be expected that the LLM sizes will continue growing exponentially. Thus, it is imperative to develop more efficient and scalable methods that allow better utilization of compute power and training data.

In this work, we present a novel approach for efficient training and finetuning, which we name **Ve**ctor projected **LoRA** (VeLoRA). Our approach is based on a key observation that the intermediate

---

†  Corresponding authors: roy.miles@huawei.com, ismail.elezi@huawei.com

38th Conference on Neural Information Processing Systems (NeurIPS 2024).

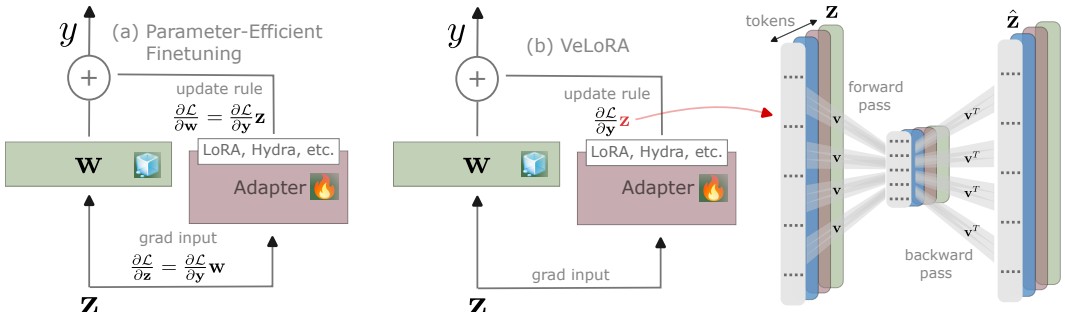

Figure 1: The memory overhead for backpropagation on a single layer consists of storing the intermediate activations and the weights. (a) demonstrates that PEFT methods can reduce the memory by using cheap low-rank adapters. (b) VeLoRA additionally compresses the saved intermediate activations to further reduce the memory usage.

activations produced during the forward propagation of deep neural networks, and kept in memory for computing the gradients during backpropagation, can be effectively represented and reconstructed from a single and fixed one-dimensional vector without losing any accuracy. This compressed representation can be made very memory efficient, with a controllable hyperparameter that provides a trade-off between the compression ratio and the model's performance. By compressing and then reconstructing the activations on the fly, our method reduces the peak activation memory footprint to a tiny fraction of what is required to store the original activations. This enables fitting much larger models into limited GPU memory compared to approaches like GaLore or gradient checkpointing.

More concretely, during the forward pass, we propose to divide each input token into a set of much smaller sub-tokens. Using a single projection vector, we then project these individual sub-tokens onto a one-dimensional subspace. Importantly, we notice that we can initialize this projection vector cheaply using first-order batch statistics and then keep it fixed throughout training. We then reconstruct the original tokens using the same vector during the backward pass. Although this reconstruction loses the original gradient properties such as the direction or magnitude, we find that it jointly encourages sparsity and locally preserves the gradient similarity, which we attribute to the overall effectiveness of the algorithm. By storing these compact representations, we can substantially reduce the memory footprint of the network during training, enabling the accommodation of larger models on hardware with limited memory capacity.

Our **contributions** are the following:

- We propose a novel compression method that reduces the memory requirement for gradient computation during training and fine-tuning of large neural network models like LLMs.

- We show that, unlike other methods, our compression method does not need expensive operations such as SVD and gradient checkpointing.

- We achieve state-of-the-art results in various benchmarks while requiring lower GPU memory compared to the baseline methods.

## 2 Related Work

**Memory-Efficient Training.** The increase in model size has necessitated the development of smart methods that make training more memory efficient. Gradient checkpointing [8] significantly lowers the memory requirements during model training by recomputing activations for the backward pass instead of storing them during the forward pass. However, doing so increases the training time from the need to re-compute gradients. Adafactor [41] and its followup [1] lowers the memory by working with the row-column outer-product factorized moments of adaptive optimizers. LOMO [30] was developed for large models and works by fusing the gradient computation and the parameter update in one step to reduce memory usage, effectively only saving the current layer gradients in memory. Recently, GaLore [54] proposed projecting the gradients onto a lower-dimensional space [9, 6], and can reduce the memory during both pre-training and finetuning. However, they store all the full intermediate activations to compute gradient updates. Their memory advantage is derived from

computing the first and second-order statistics of the gradients in a lower-dimensional space, thus limiting it to second-order optimizers only. Furthermore, GaLore needs an expensive SVD operation that introduces some significant overheads in terms of both memory and computation costs. Unlike these methods, VeLoRA does not introduce any large computation overhead while at the same time comes with a significant memory reduction. Furthermore, VeLoRA is in principle independent of the underlying optimizer.

**Parameter-Efficient Fine-Tuning (PEFT)** is an emerging field that focuses on fine-tuning a large model with a minimal number of trainable parameters. This typically involves freezing and then augmenting the original model with adapter modules. LoRA (Low-Rank Adaptation) [18] is a technique that optimizes a few rank-decomposed weight matrices during fine-tuning, rather than updating the entire set of pre-trained weights for each attention layer. This approach substantially reduces the number of trainable parameters, thereby accelerating the fine-tuning process and making it more memory-efficient. The method was later extended to also work with multi-layer perceptrons in Transformers [21, 13]. Several other methods built upon these works improving the capacity or performance of the model [21, 27, 7, 42, 35, 52, 49, 26, 45, 17]. These works can be well-complemented with quantization, further reducing the memory while keeping the performance [10, 11, 24]. Our memory-efficient algorithm is complementary to PEFT and can be used to provide additional memory efficiency in the fine-tuning regime.

**Subspace training** In [22, 15], the authors show that most of the learning process occurs within a significantly low-rank parameter space and that model weights can be effectively optimized within this lower-dimensional subspace. These subspace learning techniques have been widely adopted in various machine learning domains, including meta-learning [23] and continual learning [5]. However, unlike VeLoRA, resource-efficient training/fine-tuning is not the focus of these methods, therefore, often resulting in an overhead to meet other requirements.

**Gradient Sparsification.** Recently, there has been a surge in interest for memory-efficient training methods. In [50] only a sparse subset of the gradient vector components are stored zeroing out the remaining components. Different criteria have been proposed for selecting which gradient components to retain, such as Top-K SGD [43] which keeps only the top-k largest components, Sparsified-SGD [44] and various other sparsification methods [37, 38, 40, 25, 28, 16]. More recently, techniques combining quantization and sparsification have been proposed for resource-efficient training. Examples include TernGrad [51], Qsparse-local-SGD [3], sparse ternary compression [39], and the sparse-signSGD [32] method which combine sparsity with quantizing gradients to just the sign. A key difference is how VeLoRA compresses the intermediate activations that are used to compute gradients. Our compression algorithm is fully dense-to-dense without any pruning or sparsification of the activations. This prevents accuracy degradation issues associated with sparse updates and facilitates memory-efficient training.

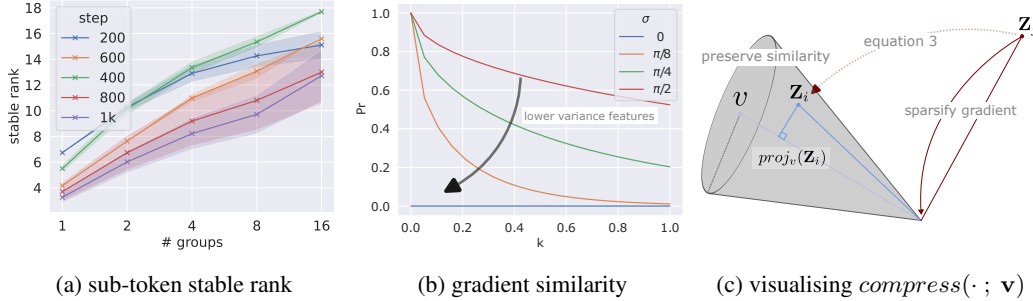

(a) sub-token stable rank   (b) gradient similarity   (c) visualising $compress(\cdot\,;\,\mathbf{v})$

Figure 2: (a) Stable rank for the input activations using a different number of groups, with $= 1$ indicating no sub-division of the tokens into smaller sub-tokens. (b) Approximate probability of the feature similarity diverging by at least $k$. (c) visualisation the rank-1 projection of sub-tokens.

## 3   Method

**The task.** In this work, we propose a memory-efficient modification to the back-propagation algorithm. Our primary motivation is to reduce the GPU memory footprint during training without

resorting to any hardware-specific quantization schemes [11] and without trading compute for memory as is done with gradient checkpointing [8].

To formalize the problem statement, let us take a step back and look at the components needed to implement back-propagation. Firstly, during the forward pass, each trainable layer in the neural network needs to store two key tensors in memory - the input activations received by that layer, and the layer's trainable weight parameters. Retaining these tensors is required for performing the parameter update rules and computing the input gradients. More specifically, during the backward pass, the previously stored input activations and model weights are used to calculate the gradients with respect to the weights and the input via the chain rule of differentiation (see Fig. 1). Storing both these sets of tensors comes with a significant memory overhead which scales with the model size. We focus on optimizing the cost of storing the input activations. We do this by compressing intermediate activation vectors and then reconstructing them when the original activations are needed for gradient calculations. This is orthogonal to PEFT [18] methods which address the overhead of saving the full-precision weights in memory by introducing cheaper trainable adapters. Further, in Section 4, we show how to combine our method with PEFT methods to achieve state-of-the-art results.

### 3.1 VeLoRA

**Overview.** Here we address the challenge of compressing intermediate activations tensors while preserving the necessary training dynamics for model convergence. Our memory-efficient algorithm consists of two components: (i) The grouping strategy to divide the original high-dimensional tokens into much smaller sub-tokens; and (ii) Fixed rank-1 projections of these sub-tokens using cheap heuristically initialized principal components. Given a large pre-trained model, we apply these steps to compress the intermediate activations saved during training while preserving most of the original model's training dynamics. We illustrate the general overview of this pipeline in Fig 1 and show PyTorch-like pseudo-code in Algorithm 1.

Consider a set of input activations that need to be saved in GPU memory during the forward pass. We denote an element in this set as $\mathbf{Z}_i = \nabla_w f(\mathbf{x}_i; w) \in \mathbb{R}^{N \times D}$, where $N$ is the number of tokens, and $D$ is the feature depth. We propose to introduce a simple grouping (reshape) operation that partitions the tokens across the depth dimension: $group(\cdot) : \mathbf{Z} \in \mathbb{R}^{B \times N \times D} \to \mathbb{R}^{B \times ND/M \times M}$ with $M$ being the new size of each token, now coined a sub-token. This operation can be described as partitioning a batch of $N$ tokens into a collection of these much smaller non-overlapping $ND/M$ sub-tokens. Then we project each of the sub-tokens onto a rank-1 subspace. The idea is that this grouping operation enables a much lower-dimensional fixed subspace to be used throughout training without any degradation in model convergence or performance. We describe the compression steps concisely as follows:

$$\mathbf{Z} \xrightarrow{group(\cdot)} \mathbf{z} \in \mathbb{R}^{B \times ND/M \times M} \xrightarrow{compress(\cdot\ ;\ \mathbf{v})} \mathbf{z}_p \in \mathbb{R}^{B \times ND/M \times 1}, \tag{1}$$

where $\mathbf{z}$ is used to denote the sub-tokens of $\mathbf{Z}$ and $\mathbf{z}_p$ are the compressed sub-tokens. This compression is achieved using the function $compress(\mathbf{z}\ ;\ \mathbf{v}) = \mathbf{z} \cdot \mathbf{v}$, which projects each sub-token onto a one-dimensional sub-space before saving them in memory. Since $M << D$ it is more memory efficient to store $\mathbf{z}_p$ instead of $\mathbf{Z}$. The initialization strategy for $\mathbf{v}$ is important for the performance of the model, however we later show that a simple and cheap average over the first batch of sub-tokens can be very effective. Finally, the compressed sub-tokens $\mathbf{z}_p$ are reconstructed for the gradient calculation during backward pass as follows:

$$\mathbf{z}_p \xrightarrow{reconstruct(\cdot\ ;\ \mathbf{v})} \hat{\mathbf{z}} \in \mathbb{R}^{B \times ND/M \times M} \xrightarrow{ungroup(\cdot)} \hat{\mathbf{Z}} \in \mathbb{R}^{B \times N \times D}, \tag{2}$$

Here $\hat{\mathbf{z}}$ and $\hat{\mathbf{Z}}$ refer to the reconstructed sub-tokens and tokens of $\mathbf{z}$ and $\mathbf{Z}$ respectively. The $reconstruct$ function projects the sub-tokens $\mathbf{z}_p$ back onto the vector $\mathbf{v}$ as a very coarse reconstruction of $\mathbf{z}$ and it is defined as $reconstruct(\mathbf{z}_p\ ;\ \mathbf{v}) = \mathbf{z}_p \cdot \mathbf{v}^T$. The overall compression and reconstruction operation is given as $proj_{\mathbf{v}}(\mathbf{z}) = (\mathbf{z} \cdot \mathbf{v}) \cdot \mathbf{v}^T$, where $\mathbf{v} \in \mathbb{R}^{M \times 1}$ is a fixed vector of unit length.

To summarize, during the forward pass, we *compress* the intermediate activation tensor $\mathbf{Z}$ into a compact representation $\mathbf{z_p}$ using $\mathbf{v}$. Then, in the backward pass when the original activation $\mathbf{Z}$ is needed for gradient computation, we reconstruct an approximation $\hat{\mathbf{Z}}$ by projecting $\mathbf{z}_p$ back onto $\mathbf{v}$.

These steps are fundamentally different and complementary to recent works that leverage the low-rank property of gradients like GaLore [54] in two ways: Firstly, they store the uncompressed intermediate activations in memory for the gradient computation. In contrast, we compress these activations explicitly during the forward pass. Secondly, GaLore relies on periodically computing the principal components with SVD to optimally capture the underlying gradient subspace. Our compression method avoids such costly operations, making it much more efficient and scalable.

## 3.2 Insights and properties of VeLoRA

**On the importance of compressing sub-tokens.** Computing the optimal low-rank subspace of the gradients using SVD is very computationally and memory intensive and often needs offloading the operation to a CPU [54]. Moreover, periodically updating the projection may be necessary to track any shift in the gradient distribution [54]. This is why dividing the tokens up into smaller sub-tokens is necessary. By doing so, it allows us to use a cheap surrogate rank-1 projective map that is initialised and frozen throughout training. Finally, one surprising observation of this grouping operation is that the sub-tokens will naturally lie on a higher-dimensional subspace than the original tokens (see figure 2a). Thus, our method cannot be faithfully explained through a better reconstruction of the gradients, but instead by a suppression of the inherently larger eigenvalues that can in turn help reduce overfitting.

**Algorithm 1** VeLoRA, Pytorch-like

```python
def forward(input, weight, v):
    # v: M x 1

    # forward compute is preserved
    out = input @ weight

    # compute vector similarity
    z = compress(group(input), v)

    save_for_backward(z, weight, v)
    return out

def backward(ctx, grad_output):
    z, weight, v = saved_tensors

    # reconstruct the input
    input = ungroup(reconstruct(z, v))

    # compute gradients
    grad_input = grad_output @ weight
    grad_weight = grad_output.T @ input

    return grad_input, grad_weight
```

**Why does a vector projection make sense?** Using a fixed vector projection throughout training fails to capture any shift in the gradients' distribution. Under the assumption that preserving the original gradient reconstruction is important, this may seem like an undesirable trait. However, quite surprisingly, we find that this shift does not hinder the model's convergence or performance at all. An explanation behind this phenomenon can be twofold: (i) The gradients become more sparse as they shift away from the initial distribution and this helps prevent the model from overfitting to the fine-tuned dataset; (ii) Although the vector projection destroys the original gradients' magnitudes and directions, it still locally preserves the gradient similarity and this similarity will govern the model's training dynamics [19].

Consider a rank-1 decomposition of two sub-tokens: $z_i$ and $z_j$. We will use the dot-product as the similarity measure $sim(\cdot)$ for which we wish to locally preserve. Let us assume that both $\mathbf{z}_i$ and $\mathbf{z}_j$ are distributed such that the angles between them and the vector $\mathbf{v}$ are normally distributed with a mean of 0 and a standard deviation $\sigma$. With a first-order approximation, the probability of the projection and reconstruction scaling the gradient similarity by at least $k$ is given as follows (see the Appendix for the full derivation):

$$Pr\left(|sim(proj_{\mathbf{v}}(\mathbf{z}_i), proj_{\mathbf{v}}(\mathbf{z}_j)) - sim(\mathbf{z}_i, \mathbf{z}_j)| > k\right) \approx 2\left(1 - \Phi\left(\frac{\sqrt{k}}{\sigma}\right)\right) \quad (3)$$

With $k > 0$ and $\sigma > 0$, this probability is bounded by $[0, 1]$. Here we can see that similarity is trivially preserved in the limit as $\sigma \to 0$. This indicates that the sub-tokens already lie on a 1-dimensional subspace spanned by $\mathbf{v}$. To further see how these gradient similarities diverge for various values of $k$ and $\sigma$, we plot equation 11 in Fig. 2c. We empirically observe that although the gradient similarity is very dependent on the distribution of features, this non-linear relationship does not hinder the model's ability to converge and generalise. Finally, we also provide an illustrative visualisation in Fig. 1 (right) that shows the locality sensitivity for preserving gradient similarity and the sparsification of gradients when they are orthogonal to the vector $\mathbf{v}$. Both of these components are important properties that we attribute to the effectiveness of VeLoRA.

**Connection to parameter efficient fine-tuning.** Although VeLoRA is complimentary to LoRA, it can indeed be viewed under the same umbrella. First let us consider LoRA, which will freeze the

original weights and only update a low-rank adapter:

$$y = Wx + ABx = (W + AB)x \tag{4}$$

Following the same analysis from FLoRA [17], we will freeze $A$ and initialise $B$ with all zeroes. i.e. $A = A_0$ and $B_0 = 0$. The weight update rule can then be given as follows:

$$\text{LoRA} \quad W' = W + A_0 \left( B_0 - \eta \frac{dL}{dB} \right) \approx \boxed{W - \eta \tilde{g} A_0 A_0^T}, \tag{5}$$

with learning rate $\eta$ and $\tilde{g} = \frac{dL}{dy} \cdot \frac{dy}{dW}$ - see FLoRA [17] for the original full derivation under the small learning rate assumption. In contrast, VeLoRA (with $M = D$ i.e. no sub-tokens) will update the original weights directly but with compressed gradients:

$$\frac{dL}{dW} \approx \frac{dL}{dy} \cdot \left( \left( \frac{dy}{dW} \cdot v \right) v^T \right) = \left( \frac{dL}{dy} \cdot \frac{dy}{dW} \right) vv^T, \tag{6}$$

which leads to the following similar weight update rule to equation 5:

$$\text{VeLoRA} \quad W' = W - \eta \frac{dL}{dW} = \boxed{W - \eta \tilde{g} vv^T} \tag{7}$$

This result highlights that VeLoRA is a special case of LoRA with a data-driven initialisation for $A_0$. Furthermore, due its construction, VeLoRA is implemented using a custom forward and backward operation rather than by modifying the architecture and fusing weights after training. Finally, **VeLoRA also provides additional compression through having a smaller shared projection $v$ for each sub-token**. This would resemble reshaping the input tensor before and after the LoRA adapter to enable smaller projection matrices.

**Cheap initialisation strategies.** GaLore [54] proposes to use the periodically updated SVD principle components to track any shifts in the underlying sub-space of the gradients. Unfortunately, this SVD operation can be very expensive both in terms of memory and compute. Another limitation is that SVD may fail to converge, and it requires casting the features back to 32-bit for numerical stability [33]. For this reason, we propose a cheap initialisation strategy. We relax the constraint on tracking the feature distribution. For all of our experiments, we use a rank-1 decomposition of sub-tokens and propose to initialize the vector **v** using the average of sub-tokens from the first batch.

## 4 Comparison with the state-of-the-art

In this section, we thoroughly evaluate VeLoRA and its individual components. In section 4.2 we demonstrate the complementary nature of VeLoRA in conjunction with many other existing PEFT methods. Section 4.4 then scales up these results to the LLaMA models, whereby we achieve a significant memory reduction when coupled with 4-bit scalar quantisation. Finally, section 4.5 extends VeLoRA to the pre-training setting where we see competitive performance alongside a real reduction for the on-device GPU memory usage.

For the VTAB-1k experiments, we applied VeLoRA to all the down projections in the trainable adapters, while for SSF we applied it to the input scale and shift layers only. For all the other experiments we simply applied VeLoRA to the value projection layer and the down projection layer of the MLP blocks.

### 4.1 Implementation details

We performed all the vision experiments on a subset of the VTAB-1k [53] benchmark for a combination of 16 different vision datasets. We finetuned a ViT-B [12] model pre-trained on ImageNet-21K using the AdamW optimizer with a learning rate of 5e-4 and a weight decay of 1e-4. We performed

Table 1: Results on a subset of the VTAB-1k benchmark. All methods use a ViT-Base-224/16 model pre-trained on ImageNet-21k. The batch sizes and ranks are the same across all tasks.

| | Memory (MB) | Natural | | | | | | | Specialized | | | | Structured | | | | | Average |
|---|---|---|---|---|---|---|---|---|---|---|---|---|---|---|---|---|---|---|
| | | Caltech101 | Cifar100 | DTD | Flower102 | Pets | SVHN | Sun397 | Camelyon | EuroSAT | Resisc45 | Retinopathy | Clevr-Count | DMLab | KITTI-Dist | sNORB-Azim | sNORB-Ele | |
| *Method* | | | | | | | | | | | | | | | | | | |
| Full tuning | 4.25 | 89.4 | 53.3 | 66.1 | 97.3 | 87.3 | 90.7 | 39.2 | 83.2 | 95.3 | 86.1 | 75.4 | 62.8 | 47.2 | 77.5 | 31.2 | 32.8 | 69.7 |
| + VeLoRA | 4.02 | 89.9 | 55.9 | 67.8 | 97.2 | 88.4 | 90.4 | 38.9 | 85.8 | 95.8 | 86.7 | 75.7 | 74.7 | 50.2 | 77.9 | 31.8 | 31.6 | 71.2 (↑ 1.5) |
| Linear probing | 1.84 | 41.6 | 86.4 | 65.9 | 97.6 | 87.2 | 36.8 | 51.1 | 79.0 | 88.4 | 72.9 | 74.0 | 34.1 | 34.8 | 59.6 | 13.2 | 22.9 | 59.1 |
| SSF | 4.13 | 89.4 | 74.0 | 72.9 | 99.2 | 91.1 | 80.7 | 56.0 | 83.3 | 94.8 | 85.3 | 75.6 | 78.5 | 45.0 | 76.9 | 23.0 | 36.9 | 72.7 |
| + VeLoRA | 3.46 | 89.1 | 74.1 | 73.0 | 99.1 | 91.3 | 80.8 | 56.3 | 82.8 | 94.9 | 85.4 | 74.8 | 78.6 | 44.7 | 75.5 | 24.6 | 36.5 | 72.6 (↓ 0.1) |
| Hydra | 3.10 | 91.3 | 72.6 | 70.9 | 99.2 | 91.3 | 88.6 | 55.7 | 82.3 | 95.2 | 85.1 | 76.1 | 81.9 | 51.7 | 78.9 | 34.5 | 40.5 | 74.7 |
| + VeLoRA | 2.88 | 91.0 | 72.8 | 70.6 | 99.2 | 91.4 | 88.2 | 56.0 | 83.2 | 94.9 | 84.3 | 75.9 | 82.7 | 51.6 | 79.9 | 34.2 | 41.4 | 74.8 (↑ 0.1) |
| LoRA | 2.86 | 89.3 | 64.7 | 68.8 | 99.1 | 90.0 | 82.3 | 52.6 | 81.7 | 95.3 | 83.7 | 74.4 | 80.4 | 47.3 | 77.9 | 28.0 | 38.1 | 72.1 |
| + VeLoRA | 2.74 | 88.9 | 67.3 | 69.6 | 99.1 | 90.7 | 83.5 | 53.3 | 81.9 | 95.2 | 83.4 | 74.3 | 79.8 | 47.1 | 78.9 | 29.7 | 40.3 | 72.7 (↑ 0.6) |

Table 2: Comparison of our method with full fine-tuning, GaLore and LORA on GLUE benchmark using pre-trained RoBERTa-Base. Our method reaches the best overall results while showing significant memory improvements, especially compared to GaLore. We bold the best results from the considered PEFT methods. The GPU memory is measured on-device.

| | Memory (GB) | CoLA | STS-B | MRPC | RTE | SST2 | MNLI | QNLI | QQP | Avg |
|---|---|---|---|---|---|---|---|---|---|---|
| Full Fine-Tuning | 4.64 | 62.24 | 90.92 | 91.30 | 79.42 | 94.57 | 87.18 | 92.33 | 92.28 | 86.28 |
| GaLore | 4.04 | 60.35 | 90.73 | **92.25** | **79.42** | 94.04 | **87.00** | **92.24** | 91.06 | 85.89 |
| LoRA | 2.71 | 61.38 | 90.57 | 91.07 | 78.70 | 92.89 | 86.82 | 92.18 | **91.29** | 85.61 |
| VeLoRA | **2.23** | **64.56** | **90.81** | 91.26 | 77.98 | **94.38** | 86.29 | 92.09 | 89.91 | **85.91** |

the language models experiments on the GLUE [48] benchmark using the encoder-decoder RoBERTa transformer [29]. We then scaled our model to large language models causal transformers using the LLama [2] family of models, finetuned in Alpaca dataset [36] and reported results on the MMLU benchmark [36]. We finally applied our method to the pre-training stage using some smaller LLama [2] models on the C4 dataset [34]. All our models were trained using the AdamW optimizer with a learning rate of 1e-3 and a weight decay of 0. We give all the other major hyper-parameters to replicate these experiments in the Appendix and also in the code.

## 4.2 Vision experiments

We conduct experiments evaluating the performance of VeLoRA for full-tuning and how it complements other PEFT methods. In Tab. 1 we reproduce a large set of results for LoRA [18], SSF [27], and Hydra [21] on a subset of the VTAB-1K benchmark, where the sub-token size for each experiment is given in the Appendix. Unlike what is more common in the PEFT literature [20, 21], we do not perform any task-specific hyperparameter tuning that would change the memory, such as batch size and rank, and to also avoid any potential overfitting to the specific task. For all experiments we used the authors provided implementations for the adapters and integrated them into the same training framework for a fair comparison. We observe that VeLoRA improves the performance compared to full-tuning by $1.5$ percentage points (*pp*), while lowering the memory requirements. We also observe that when combined with PEFT methods, VeLoRA can come with improvement in memory and performance. VeLoRA lowers the memory requirement of SSF [27] by 16% with only a minor degradation ($0.1pp$) in accuracy. It lowers the memory requirement of Hydra [21] by 7% while improving the accuracy by $0.1pp$. Finally, it lowers the memory requirements of LoRA [18] by 4% while improving the accuracy by $0.6pp$.

## 4.3 Roberta experiments

We now evaluate our method with $M = 16$ on various language tasks, using RoBERTa-Base [29] in the GLUE benchmark, and compare it with full fine-tuning, LoRA [18] and GaLore [54], presenting the results in Tab. 2. We observe that both GaLore and LoRA lower the memory requirements compared to fine-tuning from 4.64GB to 4.04GB, respectively to 2.71GB, at a cost of accuracy degradation with GaLore performance lowered by 0.39 *pp*, while LoRA accuracy drops by 0.67 *pp*.

Table 3: Mean 5-shot MMLU test accuracy for LLaMA models finetuned with adapters on Alpaca. The GPU memory estimate consists of the frozen weights, trainable adapters, and input activations.

| LLaMA Size | 7B | | 13B | | Mean |
|---|---|---|---|---|---|
| Method | Alpaca | Memory | Alpaca | Memory | |
| LoRA w/ BFloat16 | 38.4 | 8.79 | 47.2 | 15.82 | 42.8 |
| LoRA w/ Float4 | 37.2 | 5.77 | 47.3 | 9.91 | 42.3 |
| QLoRA | 39.0 | 5.77 | 47.5 | 9.91 | 43.3 |
| + VeLoRA | **39.5** | **4.88** | **48.0** | **8.48** | **43.8** |

Our method further reduces the memory needed for training to 2.23GB, an improvement of 18% compared to LoRA, and 45% compared to GaLore, while still reaching higher results than either of them. More impressively, VeLoRA reduces the memory by half compared to full fine-tuning with an accuracy degradation of only 0.37 *pp*, reaching the best tradeoff between memory and accuracy.

## 4.4 Scaling up to LLaMA

We now scale our method to large language models, demonstrating the effectiveness of VeLoRA in finetuning them. We do comparisons with LoRA on both BFloat16 and Float4, in addition to the recent method of QLoRA [11] which is widely used for fine-tuning LLMs with very low memory budget. We aim to further lower this budget, showing in the process that VeLoRA is also complementary to QLoRA, resulting in a much lower memory consumption. We present the results in Tab. 4.4 using $M = 32$ for 7B and $M = 128$ for 13B. We can see that our method outperforms QLoRA by 0.5*pp* in the Llama model, while reaching a massive performance increase compared to LoRA models. Fur-

Table 4: Comparison with low-rank algorithms on pre-training various sizes of LLaMA models on C4 dataset. Validation perplexity is reported, along with the on-device GPU memory usage.

| | 60M | 130M |
|---|---|---|
| Full-Rank | 33.52 (1.30G) | 25.08 (2.32G) |
| GaLore | 34.88 (1.27G) | 25.36 (2.02G) |
| LoRA | 34.99 (0.86G) | 33.92 (1.24G) |
| FLoRA | 34.35 (1.27G) | 25.88 (2.01G) |
| VeLoRA | **33.76** (1.18G) | **25.29** (1.83G) |
| $r/d_{model}$ | 128 / 256 | 256 / 768 |
| Training Tokens | 1.1B | 2.2B |

thermore, we reach this performance improvement, while at the same time further reducing the memory. In particular, we reduce the memory for 0.89GB, a relative improvement of 15.4% from QLoRA. We observe that this performance improvement is maintained on the larger model of 13B parameters, where again our method outperforms QLoRA by 0.5*pp* and lowers the memory requirements by 1.43GB, a relative improvement of 14.4%.

## 4.5 Pre-training on C4

We now perform an experiment where we use VeLoRA to train language models from scratch in the large C4-English dataset, presenting the results in Tab. 4. We use $M = 128$ for both the 60M and 130M, while following the same training pipeline and evaluation as GaLore and comparison with LoRA. However, unlike in the GaLore paper [54], which estimates the memory usage using the optimizer weights and memory alone, we choose to compute the real on-device memory. This quantity would take into account the cost of additionally storing the intermediate activations and also highlight the benefits of LoRA in terms of memory since the base weights will be frozen. In contrast to other experiments, our use of VeLoRA here is not with any additional adapters and is simply compressing the input activations for the original trainable base layers. We observe that our method significantly outperforms the other methods, reaching 1.08 *pp* lower perplexity than GaLore. We observe that our method outperforms GaLore in Llama-130M too.

## 5 Ablations Studies

### 5.1 Convergence properties

We observed that the rank-1 projections would encourage much higher levels of gradient sparsity (see Fig. 2b). A natural question to ask from this observation is if the gradient sparsity will come at the cost

Table 5: All three ablations are done using LLama-7B model. (a) VeLoRA has no loss in performance when trained for fewer or more training epochs than QLoRA despite both reducing the memory footprint. (b) Importance in choosing the correct number of sub-token size to find an optimal memory v.s. accuracy trade-off. Using a GPU memory estimate for the input activations only. (c) Ablating various initialisation strategies for a rank-1 projection and with $M = D / 32$.

| Epochs | QLoRA | VeLoRA |
|---|---|---|
| 1 | 36.4 | **36.7** |
| 2 | 37.3 | **37.5** |
| 3 | **38.4** | 38.1 |
| 4 | 39.1 | **39.5** |

(a) training convergence

| $M$ | Memory (MB) | Acc |
|---|---|---|
| D / 64 | 865 | 37.9 |
| D / 32 | 808 | **39.5** |
| D / 16 | 779 | 39.3 |
| D / 8 | 764 | 37.2 |

(b) sub-token size

| Method | Acc |
|---|---|
| Random | 36.8 |
| SVD | 37.1 |
| Fixed average | **39.5** |
| Running average | 38.9 |

(c) initialisation strategy

of the model's convergence. In other words, we want to verify if our model needs to be trained longer to compensate for the gradient compression. To do so, we evaluate the performance of our model at the end of each epoch and compare it with the performance of a competing model, the QLoRA. As shown in Tab. 5a, VeLoRA and QLoRA improve at the same rate. Our model outperforms QLoRA by 0.3*pp* by the end of the first epoch, and keeps this improvement rate, outperforming QLoRA by 0.4*pp* at the end of the training. In this way, we verify that the additional compression of input activations does not affect the model's convergence.

## 5.2 Sub-token size

We provide an ablation on the impact on the size of each sub-token and the model's performance, showing the results in Tab. 5b. We can see that there is a sweet spot for which the rank-1 projections of sub-tokens using an average initialisation is very effective both in terms of memory and accuracy. However, if the size of the sub-tokens is too large (i.e. when $M = D/8$), the gradients will become too sparse, which will hinder performance (see also figure 2b). In contrast, if the size of each sub-token is too small, for example with $D/64$, there is a more significant memory compression but at the cost of model performance.

## 5.3 Initialisation strategy

In Tab. 5c, we show the performance after training with various ways of initialising the vectors for each group. To avoid exceeding memory requirements we sub-sampled the tokens for SVD and we also consider the case of instance-wise initialisation. Although we would have expected better performance since the vector will always align with each incoming batch, we found that it did not lead to any performance improvement. In contrast, doing SVD initialization comes with a drop in performance. This result further confirms that the performance improvement from VeLoRA is not specifically correlated with a lower reconstruction error of the input activations.

## 5.4 Choice of layers

A key design decision for many Parameter-Efficient Fine-Tuning (PEFT) methods, including LoRA, is where to insert the adapters within the model layers. Since VeLoRA shares this core architectural choice, we aim to provide stronger intuition on VeLoRA's suitability by performing a thorough ablation study analyzing the trade-offs between memory consumption and accuracy when considering different layer placements. We present the results in Tab. 6. We observe that we achieve memory improvement in all cases where we adopt VeLoRA. However, to improve the accuracy, VeLoRA

Table 6: Memory v.s. accuracy trade-off for VeLoRA on different layers. We use a LLaMA-7B trained on alpaca and evaluated on MMLU. We report the GPU memory estimate from the input activations only.

| Query | Key | Value | Down | Memory (GB) | Acc |
|---|---|---|---|---|---|
| — none — | | | | 1.67 | 38.1 |
| ✓ | | | | 1.42 | 36.2 |
| | ✓ | | | 1.42 | 36.2 |
| | | ✓ | | 1.42 | 36.7 |
| | | | ✓ | 1.01 | 38.9 |
| ✓ | | ✓ | | 1.18 | 37.4 |
| | | ✓ | ✓ | 0.76 | **39.5** |
| ✓ | | ✓ | ✓ | 0.51 | 38.4 |
| ✓ | ✓ | ✓ | ✓ | 0.24 | 37.0 |

must be used in the MLP down-projection. A possible explanation is that this layer might suffer from overfitting on the training data or forgetting [4] from the pre-trained data. Overall, we conclude that applying VeLoRA to the value and down projection appears to be the best choice. We strengthen this claim by using this setting for all other experiments.

## 5.5 Comparison with gradient checkpointing

We compare VeLoRA to gradient checkpointing, another widely used technique to reduce the memory consumption during training. While both methods aim to minimize memory overhead, gradient checkpointing achieves this by recomputing the original activations during the backward pass, leading to a reduced memory consumption at the cost of additional compute. In contrast, VeLoRA uses a lossy compression of the activations during the forward pass and then reconstructs them in a coarser manner during the backwards, thus avoiding the need for any expensive recomputation. Our results in table 7 show that VeLoRA not only offers a comparable reduction in memory usage but also leads to faster training times compared to gradient checkpointing, as it reduces the recomputation burden and overhead.

Table 7: **On-device training time and memory costs** for pre-training LLaMA. Unlike VeLoRA, gradient checkpointing incurs a much more significant training time overhead. Batch size of 1. Our method is 17%, 30%, 30%, 47% faster than gradient checkpointing in LLama 60M, 130M, 7B and 13B. We see that the larger the model, the larger the time performance gain.

| | (a) C4 pre-training | | | | (b) Alpaca fine-tuning | | | |
| | 60M | | 130M | | 7B | | 13B | |
| Method | it/s | memory (GB) | it/s | memory (GB) | it/s | memory (GB) | it/s | memory (GB) |
|---|---|---|---|---|---|---|---|---|
| Full | 3.12 | 1.30 | 1.40 | 2.32 | 1.64 | 13.64 | 1.24 | 21.35 |
| Gradient Checkpoint | 2.47 | 1.19 | 1.03 | 1.85 | 1.14 | 7.31 | 0.78 | 11.72 |
| VeLoRA | 2.90 | 1.18 | 1.34 | 1.83 | 1.50 | 8.35 | 1.15 | 13.28 |

## 6 Conclusion

In this work, we proposed VeLoRA, a novel framework that enables the training of networks, including large language models in a highly memory-efficient manner. Our approach compresses intermediate activations during the forward pass and coarsely reconstructs them during backpropagation. VeLoRA complements PEFT methods and is able to significantly reduce memory requirements while improving the performance. VeLoRA is effective when tested in both moderately-sized vision transformers as well as in large language models. We performed experiments to demonstrate the method's effectiveness on VTAB-1K, MMLU, GLUE, and C4 benchmarks outperforming state-of-the-art methods such as LoRA, QLoRA or GaLore.

## Limitations and Broader Impact

**Limitations.** We performed all experiments on Transformer models. Although Transformers have become dominant in machine learning and computer vision, there are other important deep learning networks such as CNNs, RNNs and SSMs. It remains unclear whether our methods can be extended to non-Transformer-based models and how such an extension could be accomplished. Moreover, although our method is computationally more efficient than competing methods, its primary advantage lies in the substantial reduction of GPU memory. However, the issue of training time still persists.

**Broader Impact.** As compute power grows exponentially, model sizes grow even faster, making it challenging for smaller institutions, especially academic ones, to conduct high-quality research. This work aims to democratize AI research, particularly in large language models, by significantly reducing the memory needed for training, enabling researchers with limited compute resources to train networks and contribute to their research. However, the *democratization* of AI is controversial, with leading institutions like OpenAI, Anthropic, and Google DeepMind becoming more closed due to the potential risks of LLMs in the wrong hands. We acknowledge this concern and do not endorse misuse of our research.

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

# 7 Supplementary Material

## 7.1 Memory Estimates

In this section, we provide a detailed explanation of the memory estimates presented in the paper. For the ablation experiments in Tables 5b and 6, we only estimate the memory consumption allocated for storing the intermediate activations. We do this because all the other memory allocations (weights, optimizer states, etc.) would be the same for each experiment. The reported memory estimates will highlight the impact of each design choice or parameter on the memory of the intermediate activations, which is one of the focuses of this paper.

Additionally, for the QLoRA Llama experiments in Tab. 3, we also use the estimated memory, but we compute this from three components: (i) Saving the frozen base weights in memory, these weights are what are quantized; (ii) The trainable adapter weights, which are stored in fp16 format for all methods; and (iii) The memory cost of storing the input features for each trainable layer. We acknowledge that an efficient implementation of our method with 4-bit quantization is left for future work. However, for all other experiments using fp16, we want to highlight that we report the real on-device GPU memory usage.

## 7.2 Roberta Experiments

We fine-tune the pre-trained RoBERTa-Base model on the GLUE benchmark. For the training parameters, we follow the same settings as GaLore [54], which is given in table 8.

Table 8: Hyperparameters of fine-tuning RoBERTa base.

|  | MNLI | SST-2 | MRPC | CoLA | QNLI | QQP | RTE | STS-B |
|---|---|---|---|---|---|---|---|---|
| Batch Size | 16 | 16 | 16 | 32 | 16 | 16 | 16 | 16 |
| # Epochs | 30 | 30 | 30 | 30 | 30 | 30 | 30 | 30 |
| Learning Rate | 1E-05 | 1E-05 | 3E-05 | 3E-05 | 1E-05 | 1E-05 | 1E-05 | 1E-05 |
| Max Seq. Len. | 512 | 512 | 512 | 512 | 512 | 512 | 512 | 512 |

## 7.3 VTAB-1K Experiments

We reproduce all the reported PEFT methods under a fixed rank and batch-size setting across all tasks. This is to ensure that the memory usage is fixed for all the tasks. However, for Hydra [21] we do use the optimal scale and dropout parameters provided the authors (see table 9). The rank v.s. memory for each reported training run is provided in table 10. Here we were able to use a higher rank for LoRA to get better performance, while also reducing the total memory. We found that Hydra did not noticeably improve performance when increasing the rank and so for these we maintained the same rank for with and without the addition of VeLoRA.

Table 9: The optimal task-specific hyper-parameters proposed for the Hydra method [21].

| Parameter | Caltech101 | Cifar100 | DTD | Flower102 | Pets | SVHN | Sun397 | Camelyon | EuroSAT | Resisc45 | Retinopathy | Clevr-Count | DMLab | KITTI-Dist | sNORB-Azim | sNORB-Ele |
|---|---|---|---|---|---|---|---|---|---|---|---|---|---|---|---|---|
| Scale | 4.0 | 0.1 | 0.1 | 0.1 | 0.1 | 3.0 | 0.1 | 0.1 | 0.1 | 0.1 | 4.5 | 1.5 | 4.5 | 4.5 | 3.0 | 1.0 |
| Dropout | 0.1 | 0.2 | 0.2 | 0.0 | 0.0 | 0.0 | 0.2 | 0.2 | 0.2 | 0.2 | 0.2 | 0.2 | 0.2 | 0.2 | 0.2 | 0.2 |

## 7.4 Implementation Details

All of the experiments in sections 4.2 and 4.5 were performed using 8 NVIDIA V100 GPUs with the fp16 data type. For the LLaMA experiments in section 4.4, we trained on 4 NVIDIA A100 GPUs

Table 10: The hyper-parameters in VTAB-1K experiments. These parameters are the same for all tasks to avoid any potential overfitting to a specific task and to maintain a constant memory usage.

| Method | Rank | $M$ | Memory (GB) |
|---|---|---|---|
| Full | n/a | n/a | 4.25 |
| + VeLoRA | n/a | 32 | 4.02 |
| SSF | n/a | n/a | 4.13 |
| + VeLoRA | n/a | 32 | 3.46 |
| Hydra | 4 | n/a | 2.88 |
| + VeLoRA | 4 | 4 | 2.86 |
| LoRA | 4 | n/a | 2.86 |
| + VeLoRA | 16 | 16 | 2.74 |

using the QLoRA nf4 data type. The LLaMA experiments were based on the alpaca-lora repository [1], while the RoBERTa and C4 experiments was based on the GaLore repository [2].

## 7.5 Gradient Similarity

Using $sim(\mathbf{z}_i, \mathbf{z}_j) = |\mathbf{z}_i| \, |\mathbf{z}_j| \, cos \, \theta_{ij}$ and $proj_\mathbf{v}(\mathbf{z}_i) = |\mathbf{z}_i| \, cos \, \theta_{iv}$ we wish to see how much the gradient similarity is being preserved under various assumptions about the distributions of both $\mathbf{z}_i$ and $\mathbf{z}_j$. We can introduce a measure for this divergence as follows:

$$d(\mathbf{z}_i, \mathbf{z}_j; \mathbf{v}) = |sim(proj_\mathbf{v}(\mathbf{z}_i), \, proj_\mathbf{v}(\mathbf{z}_j)) - sim(\mathbf{z}_i, \mathbf{z}_j)| \tag{8}$$
$$= |sim\left((\mathbf{v} \cdot \mathbf{z}_i)\mathbf{v}, \, (\mathbf{v} \cdot \mathbf{z}_j)\mathbf{v}\right) - |\mathbf{z}_i| \, |\mathbf{z}_j| \, cos \, \theta_{ij}| \tag{9}$$

Since $\theta_{vv} = 0$ and $|\mathbf{v}| = 1$, we can simplify this expression as follows:

$$= ||\mathbf{z}_i||cos \, \theta_{iv}||\mathbf{z}_j||cos \, \theta_{jv}| - |\mathbf{z}_i| \, |\mathbf{z}_j| \, cos \, \theta_{ij}| \tag{10}$$
$$= ||\mathbf{z}_i||\mathbf{z}_j|(|cos \, \theta_{iv} \, cos \, \theta_{jv}| - cos \, \theta_{ij})| \tag{11}$$

Without loss in generality, we can assume $|\theta_{jv}| > |\theta_{iv}|$ and use $\theta_{ij} = \theta_{jv} - \theta_{iv}$.

$$= ||\mathbf{z}_i||\mathbf{z}_j|(|cos \, \theta_{iv} \, cos \, \theta_{jv}| - cos \, (\theta_{jv} - \theta_{iv}))| \tag{12}$$

For simplicity, consider the case where both the input activations $\mathbf{z}_i$ and $\mathbf{z}_j$ are also normalised to unit length. In practise, these magnitudes will simply be scaling the divergence linearly.

Let both $\theta_i$ and $\theta_j$ be normally distributed with standard deviation $\sigma$ and have the vector $v$ be appropriately initialised such that their means are $0$. To understand how our projection degrades the gradient similarity, we will consider the probability of $d(\cdot)$ exceeding some scalar $k$, i.e:

$$Pr(|sim(proj_v(\mathbf{z}_i), \, proj_v(\mathbf{z}_j)) - sim(\mathbf{z}_i, \mathbf{z}_j)| > k) \tag{13}$$

**Small $\sigma$ approximation**  For normally distributed $\theta_i$ and $\theta_j$, $cos(\theta_i)$ and $cos(\theta_j)$ will not be uniformly distributed and instead follow a distribution that is more concentrated around their mean values. Thus, we can use a first-order approximation of $cos(\cdot)$:

$$cos(\theta_i) \approx 1, \quad cos(\theta_j) \approx 1, \quad cos(\theta_i - \theta_j) \approx 1 - \frac{(\theta_i - \theta_j)^2}{2} \tag{14}$$

Therefore, we can express $d(\cdot)$ and $Pr(d(\cdot) > k)$ as follows:

[1] https://github.com/tloen/alpaca-lora
[2] https://github.com/jiaweizzhao/GaLore

$$d(\mathbf{z}_i, \mathbf{z}_j; \mathbf{v}) \approx \frac{(\theta_i - \theta_j)^2}{2} \tag{15}$$

$$\rightarrow Pr(d(\mathbf{z}_i, \mathbf{z}_j; \mathbf{v}) > k) \approx Pr(\frac{(\theta_i - \theta_j)^2}{2} > k) \tag{16}$$

$$= Pr((\theta_i - \theta_j)^2 > 2k) \tag{17}$$

$$= Pr(|\theta_i - \theta_j| > \sqrt{2k}) \tag{18}$$

Since $\theta_j - \theta_i$ is normally distributed with variance $2\sigma^2$ we have:

$$= 2Pr(\theta_j - \theta_i > \sqrt{2k}) \tag{19}$$

Finally, using the cumulative distribution function (CDF) of the normal distribution:

$$Pr(\theta_j - \theta_i > \sqrt{2k}) - 1 - \Phi\left(\frac{\sqrt{2k}}{\sqrt{2}\sigma}\right) = 1 - \Phi\left(\frac{\sqrt{k}}{\sigma}\right) \tag{20}$$

$$\rightarrow Pr(d(\mathbf{z}_i, \mathbf{z}_j; \mathbf{v}) > k) \approx 2\left(1 - \Phi\left(\frac{\sqrt{k}}{\sigma}\right)\right) \quad \blacksquare \tag{21}$$

