# OpenReview forum: "VeLoRA: Memory Efficient Training using Rank-1 Sub-Token Projections"
_NeurIPS.cc/2024/Conference — NeurIPS 2024 poster_

### Official Review · Reviewer_bRtV · 2024-07-07

**Soundness:** 2
**Presentation:** 3
**Contribution:** 3
**Rating:** 5
**Confidence:** 3

**Summary:**

This paper proposes a novel algorithm named VeLoRA, which achieves memory-efficient training by compressing the intermediate activations of large-scale language models into a fixed one-dimensional subspace. VeLoRA divides tokens into smaller sub-tokens and projects them during the forward pass, then reconstructs the activations during the backward pass, thereby reducing memory usage. Experimental results demonstrate that VeLoRA performs excellently across various benchmarks, complementing other parameter-efficient fine-tuning methods, and significantly reducing memory requirements while maintaining high performance.

**Strengths:**

1 The method is easy to understand and succinctly compresses the intermediate activations.

2 The method has a relatively complete derivation and explanatory analysis.

**Weaknesses:**

1 The number of parameters used by VeLoRA, especially in Tables 1 and 2, is not specified.

2 In Table 1, although there is an improvement in the average results compared to LoRA, the performance degrades on datasets such as Caltech101, Resisc45, and Clevr-Count. Why is this the case?

3 For PEFT methods, the experiments on visual tasks are insufficient. According to the category in "Yu B X B, Chang J, Wang H, et al. Visual tuning[J]. ACM Computing Surveys, 2023.", VeLoRA experiments are based only on the Parameter Tuning category, neglecting Prompt Tuning and Adapter Tuning. Therefore, the current results do not fully demonstrate the universality of the proposed method.

4 Although the authors mention time issues in the Limitations section, the training duration before and after introducing VeLoRA should be presented.

5 For Tables 1 and 2, the improvements brought by VeLoRA are not significant. Although the authors declare this in Q7 of the Checklist, a significance analysis is still necessary, especially in cases where the improvement is not evident.

**Questions:**

see Weaknesses.

**Limitations:**

The authors mention training time issues in the Limitations section, but the relevant results are not presented.

---

> ### Author Rebuttal · Authors · 2024-08-06
>
> We thank the reviewer for praising the method's $\textbf{simplicity}$, $\textbf{comprehensive derivation}$ and $\textbf{analysis}$. We are excited to see that the reviewer positively rates the contribution and presentation of the paper. We hope that our rebuttal addresses the reviewer's issues about the paper, mostly related to the training efficiency, number of parameters and method's robustness.
>
> #### **1. The number of parameters used by VeLoRA, especially in Tables 1 and 2, is not specified.**
>
> In Tables 1 and 2, we utilize the widely recognized ViT-Base-224/16 and RoBERTa-Base models. VeLoRA introduces no additional trainable parameters, **so the number of trainable parameters is exactly the same as the PEFT methods we compare with** (e.g., where we compare with LoRA, the number of trainable parameters is the same as in LoRA, same for Hydra or SSF). This is because our projections are not trainable, instead they are initialized by the batch statistics and then kept fixed.
>
>
> #### **2. In Table 1, performance degrades on specific datasets such as Caltech101, Resisc45, and Clevr-Count. Why is this the case?**
>
> For a fair evaluation, we perform a full sweep across all datasets, rather than cherry picking individual dataset results. In the cases where VeLoRA underperforms, we would like to highlight that the difference is relatively small. To emphasise this point, we have since performed two additional sweeps to report the standard deviation over all tasks.
>
> #### **3. For PEFT methods, the experiments on visual tasks are insufficient. According to the category in "Yu B X B, Chang J, Wang H, et al. Visual tuning[J]. ACM Computing Surveys, 2023.", VeLoRA experiments are based only on the Parameter Tuning category, neglecting Prompt Tuning and Adapter Tuning. Therefore, the current results do not fully demonstrate the universality of the proposed method.**
>
> We appreciate the reference to the survey paper. It's important to note that the scope of a survey paper is, by design, much broader than our submission. To demonstrate the versatility of our method, we conducted experiments across diverse settings, including image classification, small language models (RoBERTa), fine-tuning large language models (LLaMA), and pre-training LLaMA. In each scenario, our approach shows improvements in both accuracy and memory efficiency.
>
> Our work differs significantly from Prompt Tuning, which is more suited to multi-modal learning or cross-domain tasks. Our experiments focus on single modalities without employing adapters to bridge different modalities or tasks. Moreover, as noted in the LoRA paper (Section 3, "Directly Optimizing the Prompt is Hard"), optimizing for prompt tuning presents substantial challenges.
>
> Regarding adapters, we argue that the field has largely shifted towards Parameter-Efficient Fine-Tuning (PEFT) methods since the introduction of LoRA. These methods significantly outperform traditional adapter approaches without introducing additional inference overheads (as discussed in Section 3 of the LoRA paper, "Adapter Layers Introduce Inference Latency"). Therefore, in our work, we've conducted comprehensive comparisons with leading PEFT methods, including LoRA, Hydra, QLoRA, and GaLore, to ensure a thorough evaluation of our approach. We will incorporate this discussion in the final draft.
>
> Nevertheless, if the reviewer insists that we should do comparison with some particular works in the survey, we would be glad to further communicate this during the discussion period, and we will try our best to compare with them as long as the given methods provide code.
>
> #### **4. Although the authors mention time issues in the Limitations section, the training duration before and after introducing VeLoRA should be presented.**
>
> We believe there has been a misinterpretation regarding our method's computational efficiency. It's important to highlight that our approach has a similar speed as LoRA, or other comparable methods. In the limitations section, we identified two primary challenges for large models, particularly LLMs: GPU memory constraints and extended training duration (exemplified by LLaMA 3.1's training on over 15 trillion tokens).
> Our method specifically targets the memory issue, enabling researchers with limited GPU resources to train and fine-tune models. However, the inherently large datasets still require large training time. While we've made significant progress in memory efficiency, we acknowledge that reducing training time remains an area for future research, possibly through the development of more compact, information-dense datasets.
> To reiterate, **our method is faster than existing approaches such as gradient checkpointing**. We provide direct comparisons in the attached pdf to substantiate this claim. Note that our method does not increase the inference time at all, considering there are no gradients in inference.
>
> #### **5. For Tables 1 and 2, the improvements brought by VeLoRA are not significant. Although the authors declare this in Q7 of the Checklist, a significance analysis is still necessary, especially in cases where the improvement is not evident.**
>
> Thank you for highlighting this point. We've conducted additional experiments, rerunning each experiment two more times. The average accuracy of our method remains the same with a very small standard deviation of 0.1, with our method continuing to achieve the best overall performance with a consistent margin. Due to space limits we cannot show the results in the attached pdf but will put them directly in the paper.
>
> At the same time, we'd like to emphasise that the primary advantage of our approach lies in its memory efficiency, particularly for large models (as demonstrated in Tables 2-5). While the performance improvement is certainly beneficial, the method's core value is its significant reduction (sometime up to 45% compared to baseline methods and an average of 17%) in memory usage.

---

> > ### Author Response · Authors · 2024-08-11
> >
> > Dear reviewer bRtV,
> >
> > We are keen to know if there are any remaining concerns that need to be addressed or if further discussions is needed. Your feedback is invaluable in enhancing the quality of the paper and we look forward to your response. Thank you.

---

> > > ### Comment · Reviewer_bRtV · 2024-08-13
> > >
> > > Thank you for your response. It addressed most of my concerns, and I am willing to increase my score.

---

> > > > ### Author Response · Authors · 2024-08-13
> > > >
> > > > That is great to hear, thank you for your time and effort.

---

### Official Review · Reviewer_UGFV · 2024-07-11

**Soundness:** 3
**Presentation:** 3
**Contribution:** 3
**Rating:** 7
**Confidence:** 4

**Summary:**

This paper introduces VeLoRA, a novel method for memory-efficient training and fine-tuning of large language models. The key idea is to compress intermediate activations during the forward pass by projecting sub-tokens onto a fixed 1-dimensional subspace, then coarsely reconstructing them during backpropagation. This approach complements existing parameter-efficient fine-tuning (PEFT) methods and allows for significant memory savings without sacrificing model performance. The authors evaluate VeLoRA on several benchmarks including VTAB-1k, GLUE, MMLU, and C4, demonstrating competitive or improved results compared to state-of-the-art methods like LoRA, QLoRA, and GaLore while reducing memory requirements.

**Strengths:**

1.  VeLoRA presents an innovative solution to the critical problem of memory efficiency in training large language models. The method is conceptually simple yet highly effective, offering substantial memory savings without compromising performance.
2. The authors provide extensive experiments across multiple benchmarks and model sizes, from vision transformers to large language models. This thorough evaluation demonstrates the broad applicability and scalability of VeLoRA.
3. Complementary to existing methods: VeLoRA is shown to be compatible with and complementary to other PEFT methods, enhancing their memory efficiency. This makes the approach highly practical and easy to adopt in existing workflows.

**Weaknesses:**

Limited theoretical analysis: While the paper provides some intuition for why VeLoRA works, a more rigorous theoretical analysis could strengthen the understanding of its effectiveness and potential limitations.

**Questions:**

How does VeLoRA affect training and inference times compared to baseline methods? Is there a significant computational overhead?
Are there any particular types of tasks or model architectures where VeLoRA might be less effective or even detrimental to performance?

---

> ### Author Rebuttal · Authors · 2024-08-06
>
> We thank the reviewer for highlighting the $\textbf{extensive experiments, broad applicability, and scalability}$. We also appreciate your description of the approach as $\textbf{highly practical and easy to adopt}$.
>
> #### **1. Limited theoretical analysis: While the paper provides some intuition for why VeLoRA works, a more rigorous theoretical analysis could strengthen the understanding of its effectiveness and potential limitations.**
>
> We note that under “Why does a vector projection make sense?” we give a mathematical explanation, including an equality which we prove in the appendix. However, we agree with the reviewer that the paper would be even stronger with more theory. We will add further analysis to the final paper and we hope that VeLoRA will inspire other researchers to further study it theoretically, considering its very strong empirical performance, initial theoretical analysis and, potential impact in training large models, and its ease of use.
>
> We note that we showed experiments in a wide range of settings like: image classification, small language models (RoBERTa), finetuning large language models (LLaMA), and pre-training LLaMA, in all cases our method shows improvement in both accuracy and memory. VeLoRA can be adapted to any Transformer architectures using the analysis done in Sec 5.4. We leave adapting the finding of the paper to other architectures (MLP, CNN, SSM) asfuture work. Given that most SoTA models in modern AI are Transformer-based, we believe VeLoRA has significant potential for widespread impact in the field of resource efficient training.
>
> Below, we provide a theoretical justification of VeLoRA, and a better connection of it with PEFT methods, such as LoRA, which we will incorporate in the final version of the paper.
>
> #### **Theoretical connection to parameter efficient fine-tuning**
>
> We provide a theoretical analysis of VeLoRA and its similarities with LoRA. We will provide a more thorough analysis in the updated manuscript.
>
> ####  **Part 1. VeLoRA uses a low-rank data-dependant projection of the gradients**.
>
> For VeLoRA, we replace $\frac{d y}{d W}$ with a projection onto the vector $v$. For simplicity consider the case where there are **no sub-tokens**:
>
> \begin{align}
>     \frac{d y}{d W} \approx Proj_{v}\left(\frac{d y}{d W}\right) = \left(\frac{d y}{d W} \cdot v\right)v^T
> \end{align}
>
> * **Forwards.** $\frac{d y}{d W} \cdot v \in \mathbb{R}^{ND/M}$ is saved during the forward pass. These projected gradients are **much smaller** than the original gradients.
> * **Backwards.** $\left(\frac{d y}{d W} \cdot v\right) v^T \in \mathbb{R}^{ND/M \times M}$ is a reconstruction during the backward pass.
>
> This replacement modifies the gradient of the loss with respect to $W$ to:
>
> \begin{align}
>     \frac{d L}{d W} \approx \frac{d L}{d y} \cdot \left(\left(\frac{d y}{d W} \cdot v\right)v^T\right) = \left( \frac{d L}{d y} \cdot \frac{d y}{d W} \right) vv^T
> \end{align}
>
> If we denote $\frac{d L}{d y} \cdot \frac{d y}{d W}$ as $\tilde{g}$, we can express the weight update as follows:
>
> \begin{align}
>     W' = W - \eta \frac{d L}{d W} = W - \eta \tilde{g} v v^T
> \end{align}
>
> This result highlights that we are performing a data-driven rank-1 update on the matrix W.
>
> ####  **Part 2. LoRA uses a random low-rank projection of gradients**.
>
> For LoRA, we simply freeze the original weights and only train a linear low-rank adapter module that is added in parallel:
>
> \begin{align}
>     y = Wx + ABx = (W + AB)x
> \end{align}
>
> * **Frozen.** Base weights $W$ are frozen.
> * **Trainable.** Low-rank weights $A$ and $B$ are trainable.
>
> Considering freezing $A$ and initialising $B$ with all zeroes. i.e. $A = A_0$ and $B_0 = 0$. The weight update rule is given as:
>
> \begin{align}
>     W' &= W + A_0\left(B_0 - \eta \frac{d L}{d B}\right) = W - \eta \tilde{g} A_0 A_0^T,
> \end{align}
>
> with $\tilde{g} = \frac{d L}{d y} \cdot \frac{d y}{d B}$. Here we can see that VeLoRA is special case of LoRA under the lense of gradient compression. However, there are a few **important distinctions**:
>
> * **Updating base weights directly.** Since we operate in the gradient space we can update the base weights directly. Memory efficiency is achieved by only saving the projected activations in memory and using a coarse reconstruction during the backwards pass.
> * **No additional trainable parameters.** $v$ is fixed throughout training.
> * **Data-driven projections.** The vectors $v$ are initialised using the first-batch statistics, which is empirically much more effective than a random projection.
>
> ####  **Part 3.  Extended analysis to use sub-tokens.**
> Applying sub-tokening in the context of LoRA can be done as follows:
>
> \begin{align}
>     y = Wx + G_r^{-1}((A_0B)G_r(x)),
> \end{align}
>
> with $A = A_0$, $B_0 = 0$, and $G_r(\cdot)$, $G_r^{-1}(\cdot)$ being the token grouping and ungrouping operations described in the main paper. Here we can see that the proposed sub-tokening strategy motivated in the gradient space can instead be seen as another novel and more parameter efficiency PEFT method. This duality between gradient compression and parameter efficient fine-tuning is an important and a very recent emerging perspective for memory-efficient training. We will provide the more thorough analysis of these results in the updated manuscript.
>
> #### **2 .How does VeLoRA affect training and inference times compared to baseline methods? Is there a significant computational overhead?**
>
> We provide the training time (iterations/second) in Tab 1, attached pdf. Our method has a small increase in the training time compared to full-finetuning, but is still significantly faster than gradient checkpointing or other competing methods such as Galore (because of their expensive and periodic SVD). We expect that the training time can be further reduced by writing fused CUDA kernels. There is no disadvantage of our method in terms of speed during inference since there are no gradients and thus no need for any low-rank projections.

---

> > ### Author Response · Authors · 2024-08-12
> >
> > Dear reviewer UGFV,
> >
> > We are truly grateful for your positive feedback on our paper. Your constructive comments have been invaluable in helping us refine our work. We're excited to share that we have now expanded the theoretical analysis in response to your insights. Additionally, we have included a comparative speed analysis of our method against alternatives, which we believe strengthens our contribution. We hope that our rebuttal has successfully addressed your concerns. Your expertise and perspective are deeply appreciated, and we welcome any further discussion. Thank you.

---

> > > ### Comment · Reviewer_UGFV · 2024-08-12
> > >
> > > Thanks for authors' response.The response answers my questions well and theoretical analysis helps better understand the process.

---

> > > > ### Author Response · Authors · 2024-08-13
> > > >
> > > > We are glad to see that we were able to answer your questions. We will include the full theoretical analysis in the revised paper.

---

### Official Review · Reviewer_KYfS · 2024-07-12

**Soundness:** 2
**Presentation:** 2
**Contribution:** 2
**Rating:** 5
**Confidence:** 3

**Summary:**

This paper proposes VeLoRA, an activation-compression method to reduce memory consumption. VeLoRA compresses the activations by multiplying them with a vector, and the activations are then decompressed before gradient back-propagation. VeLoRA has proven to be effective on both vision and language models.

**Strengths:**

- The paper presents an interesting method to reduce memory consumption when training large models.
- Experimental results show that VeLoRA can not only reduce GPU memory, but also achieve better performance.

**Weaknesses:**

- VeLoRA reduces memory usage by compressing activations. However, the widely used gradient checkpointing method can almost entirely eliminate the need for storing intermediate activations by recomputing them. How much speed advantage does VeLoRA offer compared to gradient checkpointing?

- The rank-1 mapping method used in VeLoRA is not intuitive to me. Can you explain why the rank-1 mapping is effective from a theoretical or intuitive perspective? In my opinion, the rank-1 mapping is equivalent to a weighted average, so does this mean that setting $v$ as a constant value $ (1/\sqrt m, 1/\sqrt m, \ldots, 1/\sqrt m) \in \mathbb{R}^m $ would also be feasible?

- Although VeLoRA's reduction in memory usage is not significant, its improvement in model performance is impressive. Can you explain the reasons behind this effect?

**Questions:**

See weaknesses above.

Overall, VeLoRA is a very interesting method, but the paper should provide a more clear explanation of the intuition or insights behind the method. For this reason, I cannot definitively say that the paper is solid, but my view is generally positive.

**Limitations:**

Limitations and Broader Impact are discussed in the paper.

---

> ### Author Rebuttal · Authors · 2024-08-06
>
> We thank the reviewer for providing some constructive questions and suggestions for a more thorough and complete set of comparisons. We also appreciate the reviewer highlighting our proposed method as $\textbf{very interesting}$ and praising its $\textbf{experimental results}$.
>
> #### **1. Comparison with gradient checkpointing**
>
> While other works such as Galore or LoRA do not provide such a comparison, we completely agree with the reviewer that comparing with checkpointing should be mandatory. Thus, we provide a comparison on both the small and large-scale Llama models for pre-training and fine-tuning. See the attached PDF for the table of results, where we can see that although gradient checkpointing can significantly reduce the on-device memory usage, it comes at a relatively significant training time overhead in comparison to VeLoRA.
>
> #### **2. Uniform and normalised initialisation for v.**
>
> Projecting each sub-token onto the vector $v = (1/\sqrt{m},1/\sqrt{m},…,1/\sqrt{m})$ can work, but is much less effective than a more data-driven approach (i.e. using the first-order batch statistics). The reason for this is because we want to preserve as much information as possible when projecting the input activations. However, interestingly, we find that the best reconstruction, i.e. using SVD, does not nescasarily improve the models performance while also incurring the additional compute overheads. We have conducted an additional experiment with the $v = (1/\sqrt{m},1/\sqrt{m},…,1/\sqrt{m})$  and have updated table 5c in the main paper.
>
> See the attached PDF for the updated table with results for this initialisation strategy. Here we can see that this initialisation is around on-par with a random initialisation, but leads to a significant degradation over more data-driven initialisation strategies.
>
> #### **3. Memory usage and performance improvement.**
>
> We respectfully disagree with the reviewer that the memory reduction is not significant. The memory reduction is small only when using small networks (VTAB) and compared to LoRA (but still massive to full finetuning). However, in other cases, we get a massive memory improvement. For example, in RoBERTa experiments, our method reduces the memory compared to LoRA by 18% and 52% compared to full-finetuning. In LLaMA 7B, we reduce the memory compared to QLoRA by a further 15.5%, and in LLaMA 13B by 14.5%.
>
> With regards to the performance improvement, many PEFT methods (including LoRA) have be shown to be more effective on smaller scale tasks and datasets than full fine-tuning (such as VTAB and some sub-tasks of GLUE). With regards to the larger datasets, VeLoRA is performing an explicit regularisation on the gradient flow for the weights. More specifically, this regularisation restricts the gradient flow to follow the average (if using a batch average initialisation) from the pre-trained model. This is quite a strong regularisation that can be what is improving the models generalisation.
>
> #### **4. A more clear explanation of the intuition or insights behind the method is needed.**
>
> We give a theoretical explanation on the need for projections in section 3.2. Why does a vector projection make sense?. However, we agree with the reviewer that a more detailed explanation on the insights of the method is helpful. Thus, we make in the rebuttal, under the response of reviewer UGFV, a direct theoretical connection behind our method and LoRA, and clearly explain the gain from using the projections. In the interest of space, please see the part **Theoretical connection to parameter efficient fine-tuning** in the rebuttal addressed to reviewer UGFV.

---

> > ### Comment · Reviewer_KYfS · 2024-08-11
> >
> > Thank you for your response. My concerns have been addressed, so I will keep my score unchanged, which means I lean towards acceptance.

---

> > > ### Author Response · Authors · 2024-08-12
> > >
> > > We are happy to have addressed your concerns and happy to hear that you recommend the paper to be accepted.

---

### Official Review · Reviewer_sd52 · 2024-07-13

**Soundness:** 3
**Presentation:** 3
**Contribution:** 2
**Rating:** 5
**Confidence:** 4

**Summary:**

In this paper, the authors propose to compress the activations by down-projecting the input tensors with a vector to save memory for large-scale training. The empirical results show that their proposed method, VeLoRa, achieves better performance compared with previous work.

Strengths:

1. The topic of efficient training is important, given that the models are scaled up at a rapid rate.
2. The paper is well written and easy to follow.
3. The empirical results confirm the effectiveness of the proposed method.

Weaknesses

1. Missing several closely related work.
2. In addition to activation low-rank projection, the authors also propose a token grouping strategy. However, the benefit of this technique is unclear to me.


Given the concerns above, I recommend a borderline rejection of this paper.

[1] [Accelerating deep learning with lossy compression](https://open.library.ubc.ca/soa/cIRcle/collections/ubctheses/24/items/1.0412625) \
[2] [Flora: Low-Rank Adapters Are Secretly Gradient Compressors](https://arxiv.org/abs/2402.03293)

**Strengths:**

1. The topic of efficient training is important, given that the models are scaled up at a rapid rate.
2. The paper is well written and easy to follow.
3. The empirical results confirm the effectiveness of the proposed method.

**Weaknesses:**

1. Missing several closely related work. The most related topic is activation compression (e.g. see [1] as an example of such a line of research). In addition, the paper uses GaLore as the baseline. However, there is existing work showing superior performance without conducting expensive SVD operations (e.g. [2]). The paper lacks proper comparisons with these baselines.
2. In addition to activation low-rank projection, the authors also propose a token grouping strategy. However, the benefit of this technique is unclear to me. In Table 5 (b), the authors show that different numbers of groups will behave inconsistently: some work better while some do not without a clear pattern. It might be the variance that is playing the important role here.

**Questions:**

See the weaknesses.

**Limitations:**

The authors have adequately addressed the limitations.

---

> ### Author Rebuttal · Authors · 2024-08-06
>
> We thank the reviewer for praising the $\textbf{effectiveness}$ of our method, calling it $\textbf{important}$, $\textbf{well-written}$, and $\textbf{easy to follow}$. We also thank the reviewer for suggesting us to discuss and compare with two related works, and asking us for a clarification with regards to the token grouping strategy.
>
> ####  **1. Missing several closely related work.**
>
> We thank the reviewer for these references, which were new to us. We examined Evans' thesis and the two papers that it is based on (JPEG-ACT and AC-GC). JPEG-ACT appears less relevant as it's a CNN-specific method using JPEG compression for network features. AC-GC is more relevant, addressing activation compression, but differs significantly in its approach. It uses constrained optimization to maximize compression subject to bounded gradient error, unlike our method.
> AC-GC's experiments focus majorly on CNN and MLP networks and give no hints about their effectiveness for Transformers-based architectures which are the primary focus of our work.
> In our revised manuscript, we will incorporate citations and discussions of these works to provide appropriate context and clearly delineate the distinctions between these approaches and ours.
>
> We appreciate the author's reference to "Flora: Low-Rank Adapters Are Secretly Gradient Compressors" (ICML24), a paper we were not aware of. While published after the NeurIPS deadline, an earlier version was available on ArXiv in February 2024. Flora shares similarities with our work and GaLore in reducing GPU memory usage. However, it doesn't operate directly in activation/gradient space and, unlike our approach, doesn't save memory during backpropagation. Flora can be viewed as an alternative to adaptive optimizers (such as Adam), while VeLoRA is more of an alternative to backpropagation, making them complementary. We will acknowledge and discuss these distinctions in our revised manuscript.
>
> Furthermore, considering the relevance of the paper, we compared Flora to our method showing the results in Figure 1 (attached pdf). We ran the official code of Flora in Galore setting (having the same number of iterations as us, Galore and full-finetuning). We overperform Flora by 0.69 percentage points (pp) in Llama 60M and by 0.41 pp in Llama130M, noting that Flora reaches overall better results than Galore. Furthermore, we show in the table that we report memory improvements compared to Flora, 1.18GB vs 1.27GB in LLama 60M and 1.83GB vs 2.02GB in LLama 130M.
>
> We will update the manuscript with citation for FLoRA, the comparison and the above discussion.
>
> ####  **2. Understanding the token grouping strategy.**
>
> We appreciate the request for clarification. Projecting the input activations to a lower-dim space can be done in many ways. For example, projecting a 512-dim token to 4 dims can be achieved by:
>
> 1. Direct multiplication: Using a [512, 4] matrix, requiring 2048 parameters.
>
> 2. Grouped projection: Dividing the 512-size token into 4 sub-tokens of size 128, then projecting each sub-token to a single number. This method uses only 128 parameters, 16 times fewer than the first approach. Note that our 'parameters' are not actually trainable parameters, instead they are just initialized with batch statistics and kept fixed. Finally, we also wish to highlight that using more sub-tokens leads to an even more significant parameter (and memory) saving over a standard linear projection.
>
> We also wish to highlight that reducing the number of parameters here reduces the cost of the projection both in terms of FLOPs and memory. We have included a figure in the attached PDF to illustrate this concept visually.
>
> Not only is decomposing the token into sub-tokens important for better parameter/memory-efficiency, but it also improves the estimate of our batch statistics for initialising the projection vector $v$. Having more sub-tokens results in more samples for computing the average. Although we demonstrate strong performance for a varying number of sub-tokens (table 5b main paper), we find that careful tuning is optimal. However, we do demonstrate the robustness of VeLoRA to this hyperparameter by using the same value for all of the larger scale tasks provided.

---

> > ### Comment · Reviewer_sd52 · 2024-08-11
> >
> > Thank you for your detailed response. I raised my scores because my questions are mostly answered.

---

> > > ### Author Response · Authors · 2024-08-11
> > >
> > > Thank you and we are happy to further discuss about the paper.

---

### Author Rebuttal · Authors · 2024-08-06

We appreciate the reviewers' positive feedback on our method's **effectiveness** (sd52, KYfS, UGFV), **simplicity** (bRtV, UGFV), **comprehensive derivation and analysis** (bRtV), **writing quality** (sd52), and **thorough evaluation with comparisons to existing works** (UGFV). We're pleased that our method/topic was described as **important** (sd52), **interesting** (KYfS), and **innovative** (UGFV).
To address the reviewers' concerns, we offer the following responses:

$\\;\\;\\;$

1. **sd52**: comment on **missed related works**
    * We clarify the differences between our work and the mentioned Ph.D. thesis and ICML24 paper.
    * We provide a comparison with FLoRA (ICML24) - see attached PDF.

2. **sd52**: request for **clarification on the grouping strategy's advantages**:
    * We provide the requested explanation, and a visualization of the procedure.

$\\;\\;\\;$

3. **KYfS**: request for **comparison with gradient checkpointing**:
    * We agree this comparison is crucial and so we now provide additional results for the Llama models.
4. **KYfS**: suggestion for an additional **projection initialisation experiment**:
    * We present the requested experiment results.
5. **KYfS:** concerns about **memory improvements**:
    * We provide a detailed explanation supporting our claims.

$\\;\\;\\;$

6. **UGFV**: recommendation for **more theory** behind our method
    * We provide a theoretical connection between our method and PEFT methods such as LoRA.

7. **UGFV**: questions on **performance and potential limitations**:
    * We offer metrics on the algorithm's speed.
    * We explain that our method is applicable across all Transformer-based models, which are state-of-the-art in almost all AI subdomains.

$\\;\\;\\;$

8. **bRtV's requests**:
    * We rerun the VTAB experiments for another two times and show that our results have a very low standard deviation.
    * We emphasize that VeLoRA does **not** introduce additional training parameters compared to other PEFT methods.
    * We agree to cite the suggested survey paper.
    * Regarding experiments in other settings:
        + We explain that prompt tuning is outside our paper's scope.
        + We note that comparable works (LoRA, Hydra, Q-LoRA, Galore, Flora, etc.) don't include such experiments.
        + We argue that many other adapter methods typically underperform PEFT methods, making comparisons unnecessary.
        + We invite the reviewer to specify any particular method they'd like us to compare with, and we'll attempt to run experiments if code is available during the discussion period.

$\\;\\;\\;$

We will integrate all of these changes in the camera-ready version of the paper. We provide a figure to better illustrate the grouping strategy and tables for the new experiments in the attached pdf.

---

> ### Author Response · Authors · 2024-08-13
>
> Dear Reviewers,
>
> We would like to thank you for all the constructive comments, which are very helpful in improving our paper.
> We are glad to see that we have addressed all of the issues raised and we will incorporate these changes into the updated manuscript.
>
> Best regards,
> Authors

---

### Decision · Program_Chairs · 2024-09-25

**Decision:**

Accept (poster)

**Comment:**

The authors propose a “rank-1 on subtokens” approach to compressing activations in the forward pass and reconstructing in the backward pass. They show their method is general, performing better in pretraining and finetuning on both vision and language foundation models. I and the reviewers found the sufficiency (with justification) of a fixed projection novel and interesting. Reviewers approved of the experimental results though wanted more intuitions/comparisons/ablations/significances due to the breadth of existing memory-efficient methods, all of which the authors provided. Ultimately, all four reviewers recommend some form of acceptance with one solid Accept. **I recommend Acceptance.**